# Microsecond fingerprint stimulated Raman spectroscopic imaging by ultrafast tuning and spatial-spectral learning

Haonan Lin [1,2], Hyeon Jeong Lee [2,3,4], Nathan Tague[1], Jean-Baptiste Lugagne [1], Cheng Zong[2,3], Fengyuan Deng[2,3], Jonghyeon Shin[1], Lei Tian [3], Wilson Wong [1,5], Mary J. Dunlop [1,5] & Ji-Xin Cheng [1,2,3 ✉]

Label-free vibrational imaging by stimulated Raman scattering (SRS) provides unprecedented insight into real-time chemical distributions. Specifically, SRS in the fingerprint region (400–1800 cm$^{-1}$) can resolve multiple chemicals in a complex bio-environment. However, due to the intrinsic weak Raman cross-sections and the lack of ultrafast spectral acquisition schemes with high spectral fidelity, SRS in the fingerprint region is not viable for studying living cells or large-scale tissue samples. Here, we report a fingerprint spectroscopic SRS platform that acquires a distortion-free SRS spectrum at 10 cm$^{-1}$ spectral resolution within 20 μs using a polygon scanner. Meanwhile, we significantly improve the signal-to-noise ratio by employing a spatial-spectral residual learning network, reaching a level comparable to that with 100 times integration. Collectively, our system enables high-speed vibrational spectroscopic imaging of multiple biomolecules in samples ranging from a single live microbe to a tissue slice.

---

[1] Department of Biomedical Engineering, Boston University, Boston, MA, USA. [2] Photonics Center, Boston University, Boston, MA, USA. [3] Department of Electrical and Computer Engineering, Boston University, Boston, MA, USA. [4] College of Biomedical Engineering and Instrument Sciences, Zhejiang University, Hangzhou, PR China. [5] Biological Design Center, Boston University, Boston, MA, USA. ✉email: jxcheng@bu.edu

Stimulated Raman scattering (SRS) microscopy is a high-speed vibrational imaging modality that produces chemical maps in dynamic living systems based on intrinsic molecular vibrations[1-5]. Such capability allows direct visualization of complex biological processes without perturbation, enabling a plethora of biomedical applications, such as tracking voltage spiking during neuron firing[6], identifying the cancer margin of fresh, unprocessed tissues[7], and discovering biomarkers and therapeutic targets of aggressive cancers[8,9]. When evaluating an SRS system, speed, spectral bandwidth, and signal-to-noise ratio (SNR) are the three major aspects, which characterize the temporal resolution, chemical specificity, and reliability. Utilizing narrowband pump and Stokes lasers, single-color SRS has reached the speed of up to video-rate[10]. Meanwhile, spectroscopic SRS has been developed to acquire a Raman spectrum at each pixel, enabling the simultaneous study of chemicals with overlapping Raman bands in complex biological samples. Spectroscopic SRS has been achieved in various ways. With a broadband and narrowband laser, an SRS spectrum can be acquired by pulse shaping[11,12], spectral coding in Fourier[13-15] or compressive[16] domain, or parallel detection of a complete spectrum by a detector array[17,18]. With two broadband femtosecond pulses, spectral focusing is used to obtain an SRS spectrum by scanning the temporal delay of two linearly chirped pulses[19-22]. To date, spectroscopic SRS can record a spectrum within a ~200 cm$^{-1}$ window at the microsecond level[17,21]. Despite major advances in instrumentation that push the speed and the spectral bandwidth, most SRS applications are focused on the carbon–hydrogen (C–H) stretching region (2800–3100 cm$^{-1}$) where strong Raman bands reside. However, the highly crowded SRS signals in the C–H region severely limit the chemical specificity of SRS in a complex biological environment.

Fingerprint SRS can significantly enhance chemical specificity by providing specific Raman peaks for each biochemical component. However, two major properties of fingerprint Raman spectroscopy impose challenges to existing high-speed spectroscopic SRS implementations. First, the much weaker Raman cross-section in the fingerprint region results in a decrease in the signal level. As a means of physical compensation, one can increase the pixel dwell time, which slows down the speed and elevates the potential of photodamage. Second, the fingerprint Raman peaks for different biochemicals are spectrally narrow and close to each other. Therefore, high-spectral resolution is required to achieve high chemical specificity. Among the existing schemes, spectral focusing[23,24] is the most power-efficient since all energy of the femtosecond pulses is used. Spectral focusing SRS imaging is generally implemented by a delay stage in a frame-by-frame manner[19,20], which is not applicable to living systems. An existing high-speed spectral focusing scheme[21] by an edge-reflected resonant mirror has only 2-ps delay range, which limits the degree of chirping and leads to 28 cm$^{-1}$ spectral resolution that is not sufficient for the fingerprint region. Another galvo-based delay-line scanning setup can achieve a large tuning range and high linearity[22], but the scanning speed is limited to 1 kHz. Acousto-optical delay line has been demonstrated for pump-probe spectroscopic imaging to reach a 6-ps delay range at 34 kHz, yet the fixed delay range and scanning speed make it less versatile[25]. Polygon delay-line scanning was implemented for FT-CARS spectroscopy[26] to reach an ultrafast speed of up to 50 kHz, however, the implementation suffers from spectral nonlinearity and fixed delay tuning range, limiting its versatility and reliability. To date, a scheme that can acquire fingerprint SRS spectra at the microsecond level with a spectral resolution below 10 cm$^{-1}$ has not been reported, which prohibits spectroscopic fingerprint SRS from being a reliable tool for broad applications.

Due to the physical limits, advances of instrumentation alone are not enough to achieve reliable high-speed fingerprint spectroscopic SRS imaging. The physical limits lead to the trade-offs between speed, spectral bandwidth, and SNR, which can be conveniently expressed as a 3D hyperplane design space (Fig. 1a). Various computational methods have been proposed to extend the design space. Matrix completion[27,28] and compressed sensing[29,30] methods have been used to sub-sample images to increase speed while avoiding information loss. Denoising algorithms with models[31,32] on object structures have also been proposed to recover the SNR of images with low light exposure or low pixel dwell times. Most computational methods depend on the formulation of forward models to describe the underlying imaging process, such as the modulation of measurements by a mask, the blurring of the image by the optical point-spread function, the thermal and electronic noise of photodetector, and the laser shot noise. However, formulating a forward model requires detailed system calibration, and certain simplifications are necessary for the sake of computational tractability. In contrast, deep learning[33] offers an appealing approach that can bypass model design and directly learn features of the image to formulate mappings from raw experimental data to reliable results. In this approach, given training data of input/output image pairs, a deep neural network learns the nonlinear mappings that find optimal approximate solutions to a variety of complicated inverse problems that are challenging to address using conventional analytical methods. Deep learning has been applied to a broad range of vibrational imaging applications, such as image restoration of single-color SRS images in the C–H region under low light exposure[34] and automated detection of the tumor margin from fresh tissue[35,36]. In particular, a 3D convolution neural network (CNN) with a U-Net architecture has been successfully applied to recover the signal level of volumetric fluorescence data[37]. However, little has been done to utilize deep learning for processing spectroscopic SRS images, which are 3D image stacks with unique spectral features that are different from volumetric data. Directly applying a 3D convolution neural network (CNN) on spectroscopic data fails to treat the different physical correlations of the spatial and spectral domain, which may introduce artifacts and degrades recovery quality. Besides, a deep neural network with 3D CNN filters requires a very high computation cost and is difficult to train. Thus, a novel convolution filter designed for handling spectroscopic image datatype is much needed to facilitate deep learning as a practical tool to push the physical limits.

Herein, we demonstrate a high-fidelity fingerprint spectroscopic SRS imaging scheme with microsecond spectral acquisition speed. Such capability is enabled by integrating two innovations of ultrafast delay-line tuning by a polygon scanner and image restoration by a spatial-spectral residual net (SS-ResNet). We have developed a high-speed and high-spectral resolution spectral focusing scheme by incorporating a 55-kHz polygon scanner and a Littrow-configured reflective grating as the delay-line scanner. To compensate for the decrease in signal levels due to higher imaging speed and extensive chirping, we further apply SS-ResNet to reconstruct SRS spectroscopic images from high-speed, low-SNR raw images in the fingerprint region. We adopt a $1 \times 3 \times 3$ convolution filter on the spatial domain to capture spatial correlations and a $3 \times 1 \times 1$ convolution filter on the spectral domain to maintain spectral continuity between adjacent frames. Next, we deploy a pixel-wise least absolute shrinkage and selection operator (LASSO) regression algorithm to decompose the recovered spectroscopic image into maps of different biomolecules. The pixel-wise LASSO unmixing can effectively suppress the crosstalk between different chemical maps by incorporating the prior knowledge that at each location only a few components have dominant contributions. To demonstrate the capability of our scheme, we perform real-time imaging of lipid species,

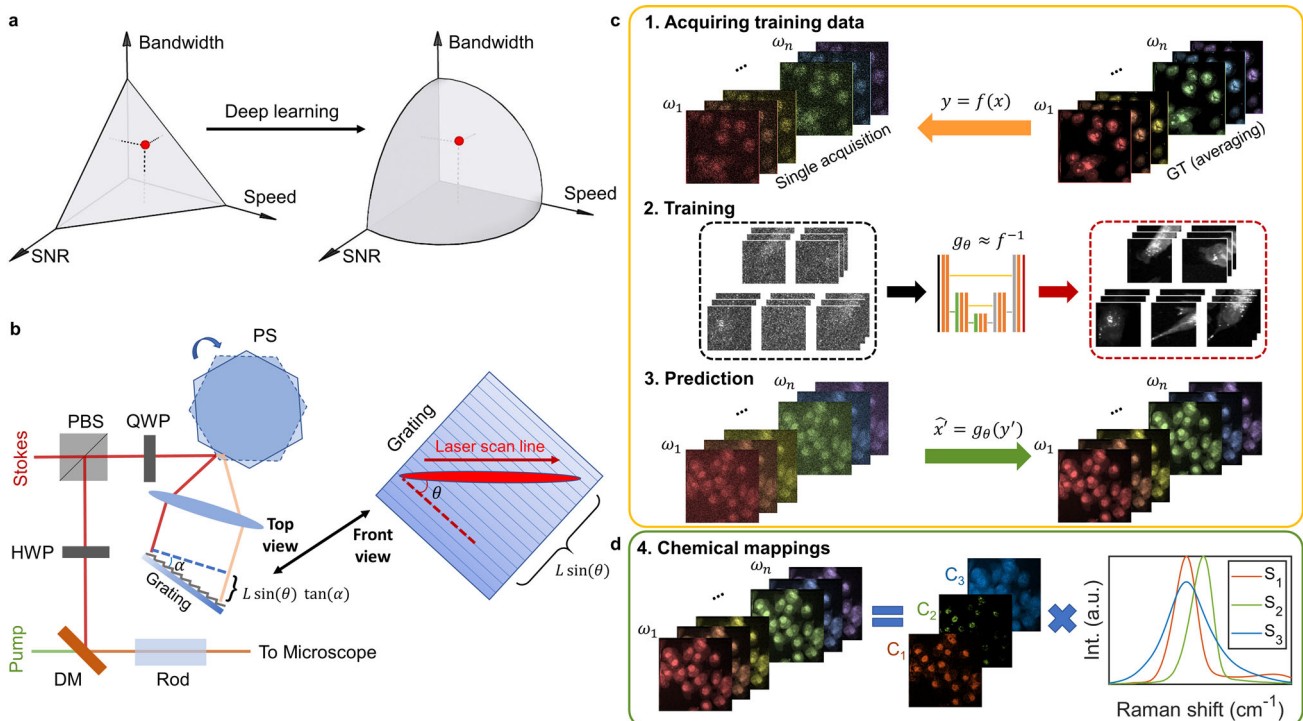

**Fig. 1 Overview of fingerprint SRS microscopy by ultrafast tuning and spatial-spectral residual learning. a** The intrinsic cross-section of the coherent Raman scattering process and instrumentation define the conventional design space for SRS imaging, resulting in trade-offs between bandwidth (i.e., spectral resolution), speed, and signal-to-noise ratio (SNR). Deep learning can expand the design space through computational methods, enabling high-speed, high-SNR fingerprint SRS imaging of living cells and large-area tissues. **b** Setup of ultrafast delay-line tuning. A 55-kHz polygon scanner is used to scan the Stokes beam onto a Littrow-configured blazed grating to generate an SRS spectrum within 20 μs. By changing the angle between the grating blazed line and the laser-scanned line ($\theta$), the effective delay range can be fine-tuned as $L\sin(\theta)\tan(\alpha)$, where $\alpha$ represents the grating blazed angle and $L$ is the length of laser scan line. PBS polarizing beam splitter, QWP quarter-wave plate, HWP half-wave plate, PS polygon scanner, DM dichroic mirror. **c** Training a spatial-spectral residual net (SS-ResNet) deep neural network for SNR improvement, ground truth (GT) images are generated by averaging multiple acquisitions of the same field-of-view, equivalent to increasing the pixel dwell time. A trained network is then applied to recover the SNR of high-speed yet noisy images. **d** Spectral unmixing using least absolute shrinkage and selection operator (LASSO) to generate chemical maps. Int., intensity. a.u., arbitrary unit.

including cholesterol and unsaturated fatty acids, in living cancer cells. Our high-speed imaging technique also allows large-area mapping of biomolecules in a whole mouse brain slice under 4 h, revealing distinctive distributions of fatty acid and cholesterol in nerve bundles and populations of cholesterol-rich cells in certain brain regions. Finally, we show the capability of differentiating multiple biomolecules by imaging biofuel production by engineered microbes. These results and applications collectively demonstrate high-speed, high-fidelity fingerprint spectroscopic SRS imaging and its potential in addressing a plethora of significant biomedical and bioengineering problems.

## Results

**Spectroscopic SRS by polygon scanner and Littrow-configured grating.** The concept of ultrafast tuning is illustrated in Fig. 1b. Briefly, two femtosecond lasers (pump and Stokes) are linearly chirped by high dispersion medium to temporally separate different frequency components. The Stokes beam is sent to a 55-kHz polygon scanner and subsequently scanned to a blazed grating at Littrow configuration, which acts as a wedge to introduce a continuous-changing path difference between the pump and the retroreflected Stokes beam. Consequently, an SRS spectrum can be acquired within 20 μs. A detailed description of the optical setup is provided in the "Methods" section and depicted in Supplementary Fig. 1a. Importantly, by rotating the blazed grating to change the angle between the laser scanning line and the grating blazed line, the effective delay range can be shortened.

Consequently, the delay tuning range is adjustable from 0 to 20 ps, which allows for extensive chirping of the lasers for dramatically improved spectral resolution. Two factors jointly determine the maximum delay range: the blazed angle of the grating and the length of the laser-scanned line, both of which are easy to change. In the current system, the long delay range enables the use of 90-cm SF57 glass rods to chirp the beams to ~5 ps, resulting in a spectral resolution of 10 cm$^{-1}$ in the fingerprint region (Supplementary Fig. 2a, b). Such spectral resolution is essential for resolving spectrally congested peaks in a fingerprint window. Also, given the linear speed of the polygon scanner, the acquired raw Raman spectrum is free of spectral channel distortion. For evaluation, we measured the spectral profiles of five chemicals and compared them with spontaneous Raman spectra (Supplementary Fig. 2c, d). Eleven significant peaks were used to map the sampling points from triggering to the Raman shifts from Raman spectroscopy, showing high linearity with $R^2 = 0.9997$ (Supplementary Fig. 2e). The sensitivity was quantified by acquiring SRS spectra from dimethyl sulfoxide (DMSO) diluted with DI water (Supplementary Fig. 2f). Besides the background due to cross-phase modulation, the DMSO solutions contributed to a significant peak at 2913 cm$^{-1}$. At concentrations as low as 0.125% v/v, the DMSO peak was still separable from the background, suggesting a high sensitivity in the C–H region. However, in the fingerprint region, excessive averaging is necessary to obtain an SRS spectrum with high SNR (Supplementary Fig. 2g, h).

**SNR recovery via spectra-spatial residual net and chemical mapping by pixel-wise LASSO unmixing**. To extract information from the high-speed yet noisy spectroscopic images, we apply a two-step processing approach that involves SNR recovery and chemical mapping. To recover the SNR, we deployed a deep neural network, acting as a supervised denoiser, to recover the SNR of high-speed fingerprint SRS images (Fig. 1c). We first generated pairs of spectroscopic SRS images as the training set, with high-speed, low-SNR images as the raw acquisition and a low-speed, high-SNR image (through averaging of multiple raw acquisitions) as the ground truth. The framework of the network is shown in Supplementary Fig. 3a. Due to the large size of each spectroscopic image and the experimental difficulty of generating a large number of training images, the network is based on the U-net[38] encoder-decoder structure. The up-sampling and skip-connection layers in the network improves the resolution of learned features and thus requires less training samples. Inspired by the pseudo-3D network for video processing[39], we consider the different physical correlations between spatial and spectral domains and replaced the widely used $3 \times 3 \times 3$ 3D CNN filter with two parallel filters, including a $1 \times 3 \times 3$ spatial convolution filter and a $3 \times 1 \times 1$ spectral convolution filter to maintain spectral continuity between adjacent frames (Supplementary Fig. 3a). Each convolution filter is detailed in Supplementary Fig. 3b. Since the memory cost for each filter is reduced, our network could incorporate six filters at each layer without exceeding the GPU memory limit. Finally, a residual learning scheme[40] is applied to facilitate the training of a deep network. The overall U-net structure greatly reduced the need for the number of training samples.

After SNR recovery, the spectroscopic image stack is linearly decomposed into chemical maps (Fig. 1d) to facilitate downstream visualization and analysis. Based on the observation that at each spatial location, only a few chemical components have dominant contributions, we used pixel-wise LASSO regression[41] to incorporate individual $L_1$-norm sparsity regularization to the concentrations at each pixel. The level of regularization can be fine-tuned such that the output can suppress crosstalks between different channels while avoiding artifacts. We applied the approach to unprecedented imaging conditions reaching high speed, high SNR, and high chemical specificity in the fingerprint region for a wide variety of biological samples. The applications include living cancer cells, whole mouse brain slice, and single bacteria, with a focus on chemicals that are difficult to study in the C–H region.

**High-speed spectroscopic fingerprint SRS imaging of lipid metabolism in Mia PaCa-2 cells**. Lipid metabolism is a cellular process involving spatiotemporal dynamics of fatty acid and cholesterol. The distributions of different lipid species in the cell are tightly regulated to ensure proper cellular activities and function. Abnormal lipid metabolism is related to many human diseases, including aggressive cancer[8,9]. Thus, quantitative imaging of lipids in living systems is of great interest. Unlike fluorescence imaging of lipophilic dyes, Raman spectroscopy provides high chemical specificity to differentiate lipid species, such as cholesterol and various fatty acids. With enhanced signal levels, SRS is capable of quantitative imaging of specific lipid species. For example, cholesterol mapping has been demonstrated in cholesterol-rich samples such as the atherosclerotic artery[42] and lysosome-related organelles in *Caenorhabditis elegans*[43] by focusing on the sterol C=C stretching band at 1669 cm$^{-1}$. However, due to the limited signal levels in the fingerprint region, except in the abovementioned cases of excessive accumulation, it remains challenging to study cholesterol in single living cells or large-area tissues.

To demonstrate that our system can achieve real-time lipid tracking in living cells, we imaged Mia PaCa-2, an aggressive pancreatic cell line, within the 1550–1750 cm$^{-1}$ fingerprint vibrational window. For training, we first acquired a dataset consisting of pairs of raw and ground truth images of Mia PaCa-2 cells. We used fixed Mia PaCa-2 cells to ensure that the ground truth images formulated by excessive averaging do not suffer from motion artifacts. Each raw spectroscopic image stack covering a ~200 cm$^{-1}$ spectral window with $200 \times 200 \ \mu m^2$ field-of-view (FOV) was acquired within 1.8 s. A total of ~20 image pairs were used for training. For the raw images, the SNR of a cell region at 1650 cm$^{-1}$ was ~1.4. The ground truth image was generated by averaging 100 raw images of the same FOV, resulting in a ~10-fold SNR enhancement. After training, the performance of SNR recovery was validated using a set of previously unseen images. We compared the raw, network-recovered, and ground truth images of the same FOV at 1650 cm$^{-1}$ (Fig. 2a–c), demonstrating that the SS-ResNet recovery allows reconstruction of the raw spectroscopic image stack, reaching comparable image quality to the ground truth images. The same validation dataset was processed by block-matching 4D filtering (BM4D)[44], a state-of-the-art unsupervised 3D image denoising algorithm. Also, to compare the performance of spatial-spectral convolution, a U-net with 3D CNN was trained and tested on the same dataset. The results (Supplementary Fig. 4a–e) suggest that both networks outperformed BM4D significantly. Meanwhile, the SS-ResNet is better than 3D CNN by maintaining more detailed structures without introducing artifacts. We further quantified the observations by calculating the normalized root mean square error (NRMSE) and structural similarity (SSIM) index[45] for the raw vs. ground truth, BM4D vs. ground truth, 3D CNN vs. ground truth, and SS-ResNet vs. ground truth (Supplementary Fig. 4f, g). Both measurements suggest significant improvement of the image quality using SS-ResNet. The averaged spectral profiles from a small region of interest for the raw, ground truth, and recovered images (Supplementary Fig. 5) show that the network introduces no spectral distortion. To test whether the network recovery facilitates downstream spectral analysis, we selected a small region of interest from the validation set (Fig. 2d) and performed pixel-wise LASSO unmixing on raw, SS-ResNet, and ground truth image stacks. We used three SRS spectral profiles generated from bovine serum albumin (BSA), triglyceride, and cholesterol (Fig. 2e) as the references. These spectral profiles represent three major Raman bands, namely the amide I band at ~1650 cm$^{-1}$ from proteins, the acyl C=C band from lipid acyl chains at 1650 cm$^{-1}$, and the sterol C=C band from cholesterol a 1669 cm$^{-1}$. The outputs from the network and the ground truth show similar spatial distributions and concentrations for all three components (Fig. 2f). In contrast, the results from the raw data failed to provide insights into the distributions of chemical species and were difficult to distinguish from the background noise (Fig. 2f). To quantify the quality of chemical maps after SNR recovery, we calculated the SSIM index for all the three chemical channels (Fig. 2g). The SSIM indices increased considerably after recovery, which proved that our approach did not introduce artifacts and provided reliable results on the subsequent chemical analysis.

To apply this high-speed, high-sensitivity method to the real-time mapping of lipid in living cells, we imaged living Mia PaCa-2 cells and recovered high-resolution images from the raw images taken at high speed by applying the same SS-ResNet trained on fixed cells. In living Mia PaCa-2 cells, lipid droplets are shown to be highly dynamic[46]. Live-cell imaging at the speed of 1.8 s per stack was performed on Mia PaCa-2 cells to capture lipid droplet dynamics. In contrast, we observed severe motion artifacts in the 100-averaged image from the live-cell data (Fig. 2h). SS-ResNet recovered images from a single frame showed clear circular-

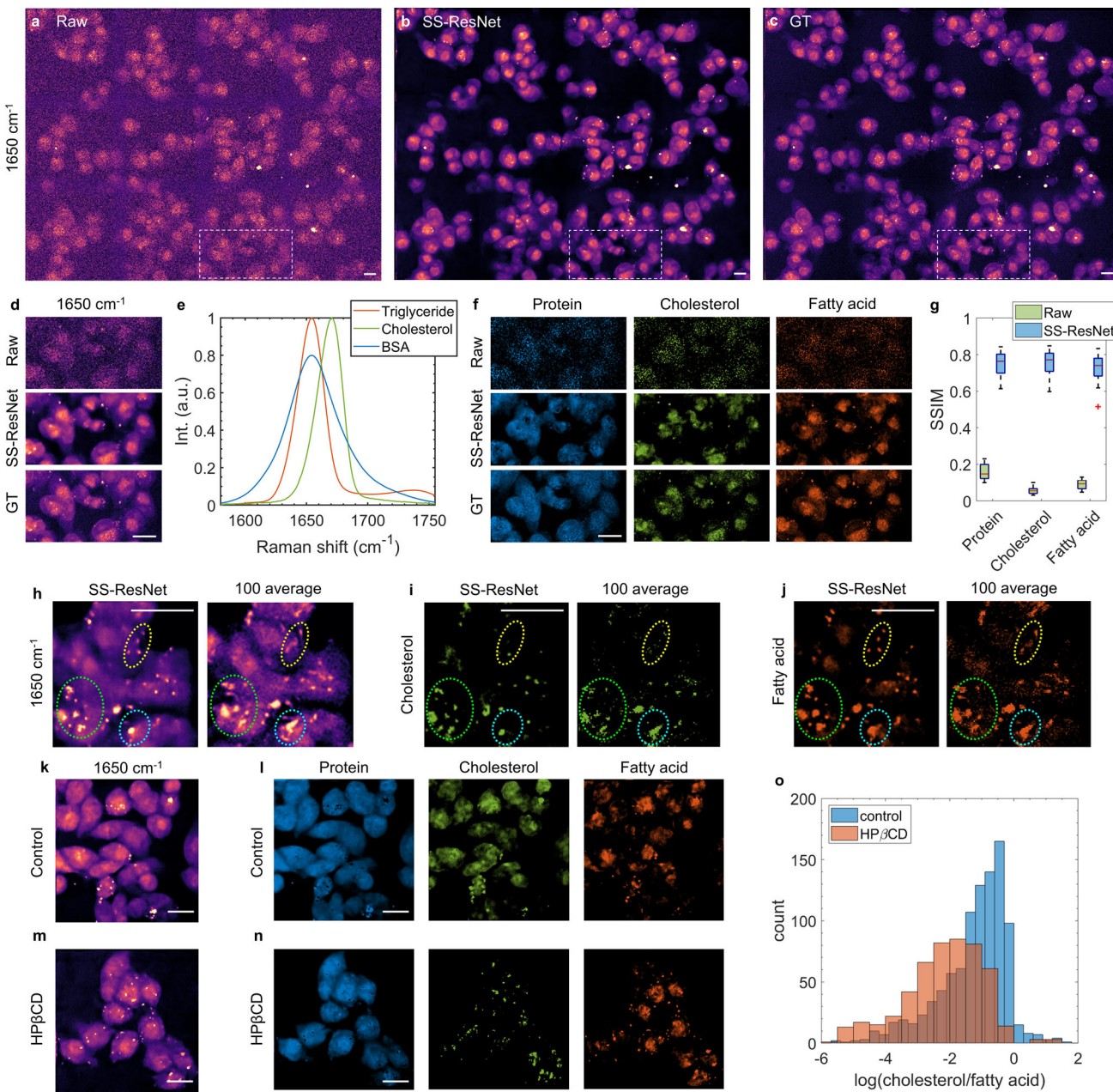

**Fig. 2 High-speed spectroscopic fingerprint SRS imaging of living Mia PaCa-2 cells. a–c** Fingerprint spectroscopic SRS imaging of fixed Mia PaCa-2 cells by single raw acquisition, spatial-spectral residual net (SS-ResNet) recovery from raw data, and 100 images averaging ground truth (GT). **d** Zoom-in comparison of the same region of interest shown in white dashed boxes in (**a–c**). **e** Fingerprint SRS spectra of bovine serum albumin (BSA), cholesterol, and triglyceride serving as the pure spectral references of protein, cholesterol, and unsaturated fatty acid. Int., intensity. a.u., arbitrary unit. **f** Chemical maps of protein, cholesterol, and fatty acid by pixel-wise LASSO unmixing from spectroscopic SRS images shown in (**d**). **g** Quantitative analysis of chemical mapping accuracy after network recovery. Box plots ($n = 19$) show the SSIM for raw vs. GT and network vs. GT of the three chemical channels. The boxes show interquartile range (IQR), the red line indicates medians, the black lines represent whiskers which extend to 1.5 times of the IQR and the red data points are the outliers exceeding the whiskers. SSIM, structural similarity index. **h** Fingerprint spectroscopic SRS imaging of living Mia PaCa-2 cells by SS-ResNet recovery of single raw acquisition and 100 averaging. **i, j** Cholesterol and fatty acid maps by LASSO unmixing from data in (**h**). Three significant motion artifacts are highlighted as circled regions. **k, l** High-speed imaging of living Mia PaCa-2 cells in normal conditions after network recovery, followed by chemical maps of protein, cholesterol, and fatty acid. **m, n** High-speed imaging of living Mia PaCa-2 cells with HPβCD treatment after network recovery, followed by chemical maps of protein, cholesterol, and fatty acid. **o** Single-cell statistical analysis of the ratio between cholesterol and fatty acid over ~1000 cells in control and HPβCD-treated group. Scale bars, 20 μm.

shaped droplets within the cells, highlighting the importance of temporal resolution during live-cell imaging. The chemical maps of cholesterol and fatty acid (Fig. 2i, j) further confirmed that motion artifacts affect the fidelity of the subsequent spectral analysis. After recovery, clear lipid dynamics can be visualized at

1650 cm$^{-1}$ (Supplementary Videos 1–2), and real-time chemical mapping of protein, cholesterol, and fatty acid can be achieved (Supplementary Videos 3–5).

We further asked whether this method could be used to track changes in cholesterol amount and distribution. To that end, we

imaged two sets of living Mia PaCa-2 cells, a control set and a set treated with HPβCD, which extracts cholesterol from the cell membrane[47]. Compared with the control group, the cholesterol concentration in the cell membrane decreased significantly after HPβCD treatment, whereas the fatty acid concentration maintained at the same level (Fig. 2k–n). The remaining cholesterol after HPβCD treatment is mainly distributed within the lipid droplets (Fig. 2n). By calculating the single-cell ratio between cholesterol and fatty acid concentrations for ~1000 cells from the control and the HPβCD-treated groups, we confirmed significant reductions in cellular cholesterol after the treatment (Fig. 2o). These data collectively show that deep-learning high-speed fingerprint SRS imaging enables high-fidelity, real-time chemical mappings of chemical bonds in single living cells and facilitates the tracking of metabolite dynamics at subcellular levels.

**Chemical mapping of a whole mouse brain slice by high-throughput spectroscopic fingerprint SRS imaging.** Brain tissue is comprised of many cell types, and biomolecules in the tissue are highly heterogeneous among different brain areas. Chemical mapping of the whole brain is essential for studying the functionality of molecules in the brain. Previous label-free metabolic studies of mouse whole-brain slices were mainly based on multi-color SRS imaging in the C–H window, providing only protein and lipid information[7,48]. For the sake of maintaining sample conditions during the experiment, the total acquisition time of a mouse whole-brain slice is usually several hours. Therefore, it remains challenging to perform spectroscopic SRS imaging in the fingerprint region to generate chemical maps of other biomolecules.

Here, we used a fixed mouse brain frozen-sectioned to 150-μm slices as the testing sample. Following the procedures in Fig. 1c, we first took a training and validation dataset in different brain regions, including the lateral hypothalamus (LH), caudate putamen (CPu), cortex (CTX), habenula (HB), medial habenula (MH), ventral lateral nucleus (VL), hippocampus (HC), dentate gyrus (DG), and corpus callosum (CC). Due to the much-complicated spatial features in the brain tissue, a total of 50 training image pairs were taken for training. Each raw image was taken at a speed of 3.8 s per spectroscopic stack with a 100 × 100 μm² field-of-view (FOV), resulting in ~2.4 SNR level. The high-SNR ground truth GT image was acquired by averaging the raw measurements of the same FOV 100 times (Supplementary Fig. 6). After training, a validation set was used to test the ability to recover SNR using SS-ResNet. After recovery, the SNR of the raw image improved significantly with the subcellular details preserved, reaching comparable image quality to the ground truth image (Fig. 3a–c). To quantify the reconstruction quality, we measured the NRMSE and SSIM for the raw vs. ground truth and SS-ResNet vs. ground truth for 15 different validation images (Fig. 3d). A comparison between the performance of SS-ResNet, 3D CNN, and BM4D is illustrated in Supplementary Fig. 7. Like the previous dataset, the performance of SS-ResNet is significantly better than BM4D and slightly outperforms 3D CNN. The averaged spectral profiles from a selected region of interest for the raw, GT, and recovered images are shown in Supplementary Fig. 8. Taking advantage of the high imaging speed of our system and the ability to recover high SNR by SS-ResNet, we performed fingerprint SRS spectroscopic imaging of a mouse whole-brain slice. Acquisition of the whole-brain slice over a ~200 cm⁻¹ spectral window in the fingerprint region was finished within 3.5 h, which is comparable to the acquisition time of multi-color SRS imaging in the C–H region focusing on a few Raman shifts[7]. The comparison between the raw image and the network recovered the image of the whole-brain tissue at 1650 cm⁻¹

demonstrates that morphologies of single cells and nerve bundles within the brain can be clearly distinguished after recovery (Supplementary Fig. 9).

We further applied pixel-wise LASSO spectral analysis of the SS-ResNet recovered image stack to produce chemical maps of the amide I group (blue, for protein), acyl C=C (red, for unsaturated fatty acid), and sterol C=C (green, for cholesterol) for the whole-brain slice (Fig. 3e–g). The composite image of the three components shows significant heterogeneity among different cells and brain structures (Fig. 3h), reflecting a relative abundance of protein, fatty acid, and cholesterol. To further characterize the distribution of the biomolecules, we focused on several brain regions and features (Fig. 3i). Overall, the soma of mature neurons shows relatively lower concentrations of all three components compared to the surrounding tissue. Surprisingly, we found abundant cholesterol-rich cells present near neurons in the LH and basal amygdaloid (BM) regions, which may represent different metabolic activities in this population of cells. We also observed that nerve bundles in the ventral posterior nucleus (VP) and CPu are comprised of different ratios of cholesterol and fatty acid. Interestingly, there are a few rare cells that contain high cholesterol concentrations in the DG region (Circled regions in Fig. 3i). As DG is one of the regions containing neural stem/progenitor cells, we suspect that these cholesterol-rich cells may reflect cells undergoing hippocampal neurogenesis. In summary, large-area SRS imaging in the fingerprint region is a viable tool for label-free measurement of cellular cholesterol content, which could be used to address the relationship between cholesterol metabolic activity and a variety of brain diseases and disorders, including various neurodegenerating disorders and brain tumors.

**High-throughput spectroscopic fingerprint SRS imaging of *E. coli* biofuel production.** Limonene and pinene are biofuel precursors that can be produced biosynthetically in microbes such as *Escherichia coli* (*E. coli*) using strains that have been engineered to produce the enzymes necessary to synthesize these chemicals[49–51]. Currently, quantitation of biochemical production levels mainly relies on gas chromatography-mass spectrometry (GC–MS), which suffers from low throughput and requires extraction steps that destroy the sample. Strain engineering and optimization typically involve the construction of many variants, followed by screening, in a lengthy iterative process. The limited throughput of GC–MS approaches hinders efficient optimization of design variables for biochemical synthesis. In addition, GC–MS only provides quantification of population-level production, ignoring the potential for genetic or phenotypic variation among cells[52,53]. Thus, a high-throughput quantification method that provides direct measurement of biofuel concentrations has the potential to improve the design, build, and test cycle necessary for improving production strains. SRS is a promising approach to fulfill this requirement by detecting intrinsic vibrational signatures from the biofuels. Yet, due to the overwhelming SRS contributions from endogenous proteins and lipids, quantitative imaging of the production levels for certain biofuels (i.e., limonene, pinene) in the crowded C–H region has not been reported. High-throughput SRS imaging in the fingerprint region is expected to address this challenge by providing specific and well-separated Raman spectra for the biofuels.

We have applied our platform to perform high-throughput quantitative chemical imaging of chemical compounds produced by genetically engineered *E. coli*. Figure 4a depicts SRS spectra in the ~1650 cm⁻¹ fingerprint Raman window, in which unsaturated fatty acid contributes to the peak at 1655 cm⁻¹, limonene has two peaks at 1645 and 1678 cm⁻¹, while α-pinene contains a peak at 1660 cm⁻¹. The peaks all originate from C=C bonds but

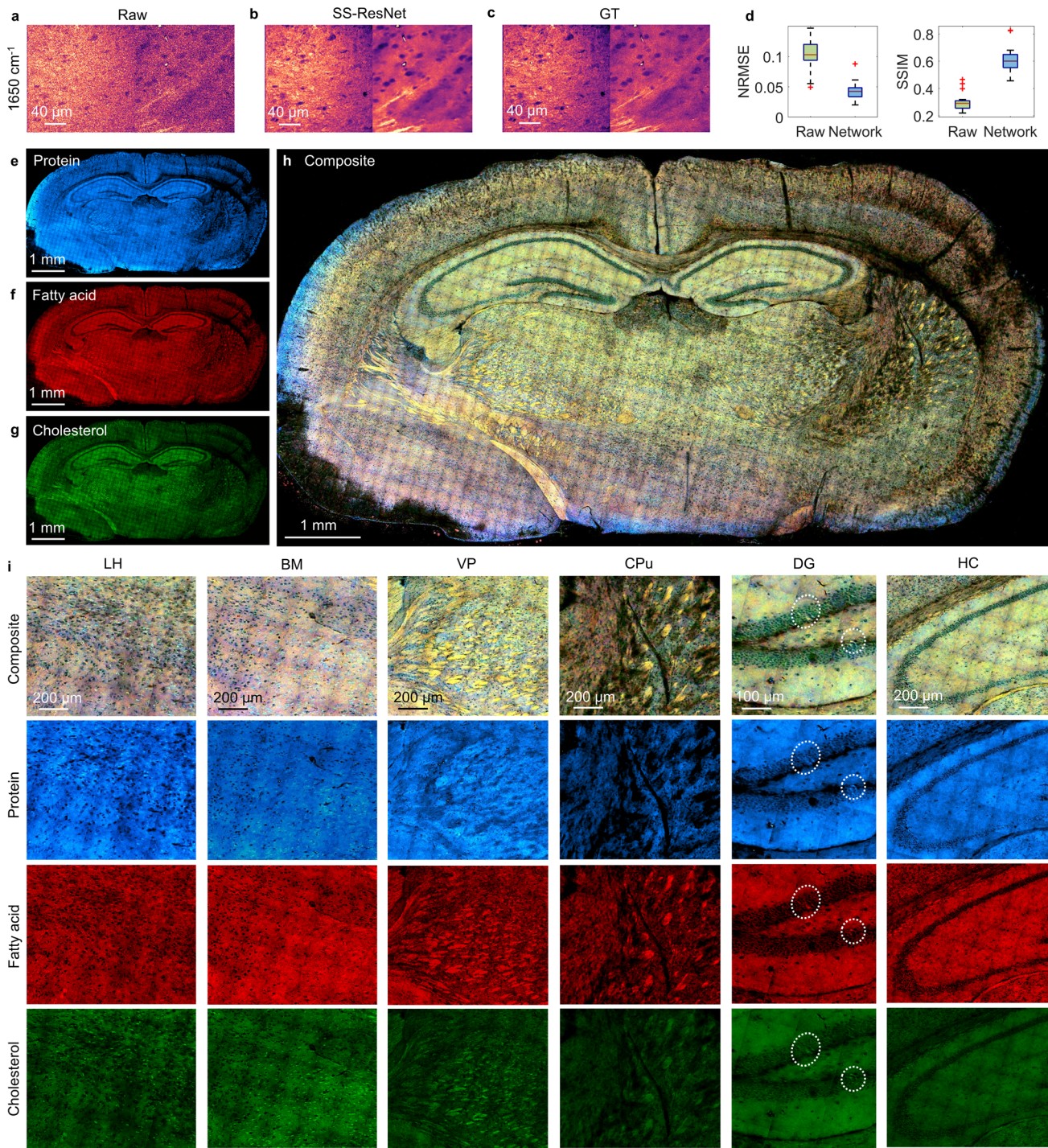

**Fig. 3 Spectroscopic fingerprint SRS imaging of mouse whole brain. a–c** Fingerprint spectroscopic SRS imaging of mouse brain by single raw acquisition, network recovery, and 100 averaging ground truth (GT). **d** Quantitative analysis of network recovery quality for mouse brain. Box plots ($n = 15$) show the NRMSE and SSIM for raw vs. GT and network vs. GT. The boxes show interquartile range (IQR), the red line indicates medians, the black lines represent whiskers which extend to 1.5 times of the IQR and the red data points are the outliers exceeding the whiskers. NRMSE normalized root mean square error, SSIM structural similarity index. **e–g** Protein, fatty acid, and cholesterol maps of a mouse whole-brain slice after SS-ResNet recovery. **h** Mouse whole-brain composite chemical maps consisting of protein (blue), fatty acid (red), and cholesterol (green). Different colors indicate different percentage concentrations from the three channels. **i** Zoom-in images of different mouse brain areas. Circled regions in the DG area include rare cells with high cholesterol content. LH lateral hypothalamus, BM basal amygdaloid, VP ventral posterior nucleus, CPu caudate putamen, DG dentate gyrus, HC hippocampus.

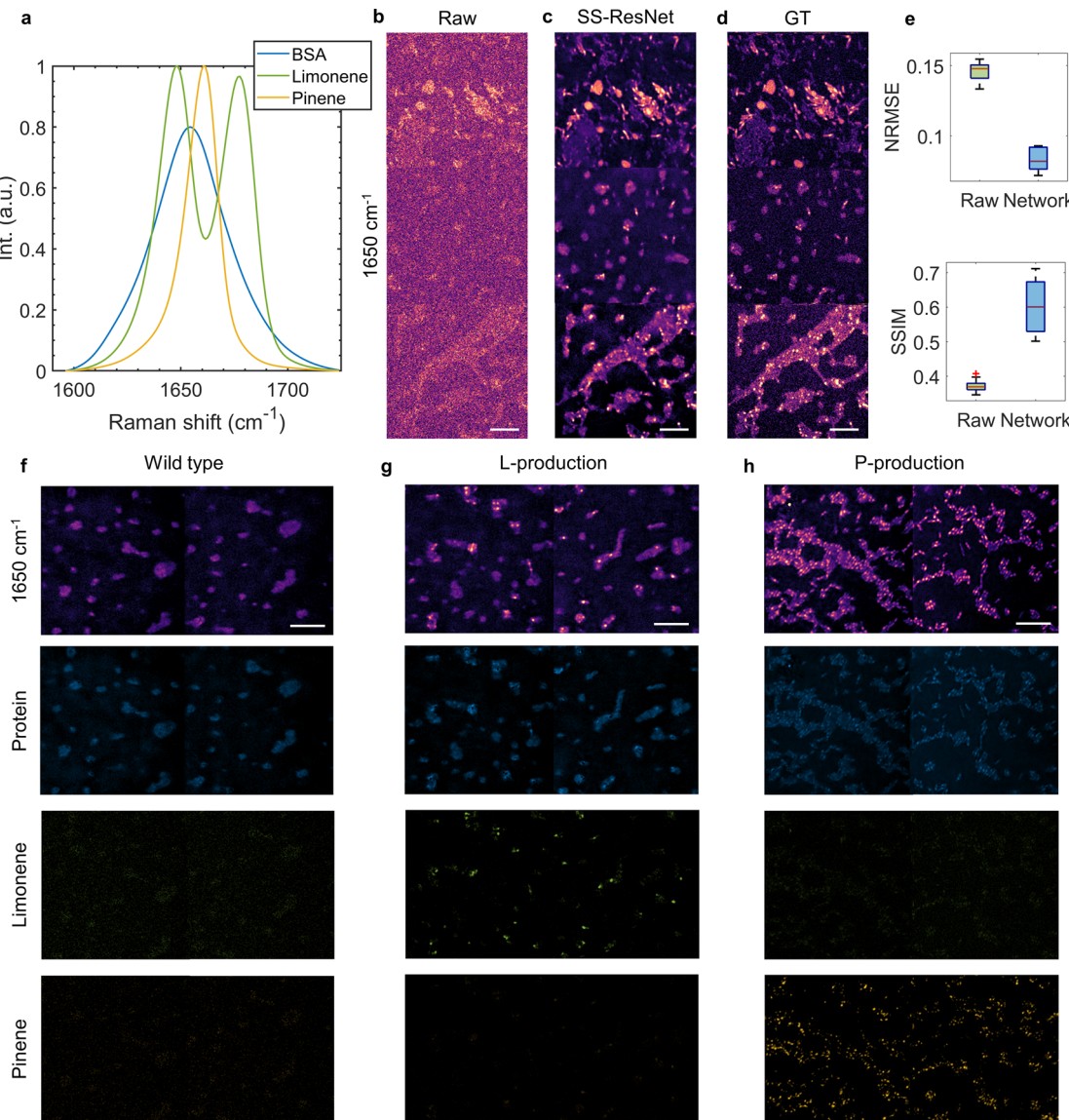

**Fig. 4 High-speed imaging of *E. coli* biofuel production strains. a** Fingerprint SRS spectra of bovine serum albumin (BSA), pure limonene, and pinene samples. Int., intensity. a.u., arbitrary unit. **b–d** Fingerprint spectroscopic SRS imaging of *E. coli* by single raw acquisition, spatial-spectral residual net (SS-ResNet) recovery from raw data, and 100 averaging ground truth (GT). **e** Quantitative analysis of network recovery quality for the *E. coli* dataset. Box plots ($n = 13$) show the NRMSE and SSIM for raw vs. GT and network vs. GT. The boxes show interquartile range (IQR), the red line indicates medians, the black lines represent whiskers which extend to 1.5 times of the IQR and the red data points are the outliers exceeding the whiskers. NRMSE normalized root mean square error, SSIM structural similarity index. **f–h** Network-recovered image at 1650 cm$^{-1}$ and chemical mappings of protein, limonene, and pinene for wild type, limonene production, and pinene-production strains. L-production, Limonene production. P-production, Pinene production. Scale bars, 10 μm.

differ from each other due to the specific structures of each chemical. In addition, the amide I group from protein contributes a broad Raman band around 1650 cm$^{-1}$, serving as the contrast for the cell body. Using the same strategy as in the previous two applications, we acquired training and testing sets from both wild-type cells and biofuel production strains (~20 image pairs in total), which consisted of pairs of high-speed, low SNR (~1.8) and low-speed, high-SNR GT images through 100 averages. After training, SS-ResNet was applied to a validation set to test the recovery performance. Examples of validation images at 1650 cm$^{-1}$, including the raw image, SS-ResNet recovery, and ground truth, are shown in Fig. 4b–d. Further quantitation of the reconstruction quality is depicted in Fig. 4e, suggesting that it is possible to denoise images while maintaining high-quality spatial localization data. Further comparison with 3D CNN and BM4D is shown in Supplementary Fig. 10. As

depicted in Supplementary Fig. 11, the spectral quality of the data is maintained after denoising.

Next, we performed high-speed imaging and SS-ResNet recovery on images of a wild-type *E. coli* strain, which does not produce biofuel. We compared this to limonene production[50] and pinene-production[51] strains of *E. coli*. Based on the spectral profiles from pure chemicals, pixel-wise LASSO spectral analysis decomposed the network-recovered spectroscopic images of the strains into the maps of the three chemicals. The chemical maps indicated that the wild-type strain (Fig. 4f) only had significant signals from the protein in the cell bodies, whereas the limonene (Fig. 4g) and pinene (Fig. 4h) producing strains had protein signals and a substantial increase in the corresponding concentrations of intracellularly aggregated chemicals. We did not include fatty acid and cholesterol in the analysis due to the negligible contributions. Using our scheme, the acquisition time

of fingerprint SRS imaging was 8 s for a $50 \times 50 \, \mu m^2$ FOV covering hundreds of *E. coli* cells, offering excellent potential for high-throughput screening to optimize the design variables of biofuel production pathways. An independent verification of the production level was performed by measuring the biofuel concentrations of the whole culture using GC–MS (Supplementary Fig. 10). From the GC–MS results, limonene and pinene are clearly present in the limonene and pinene-production strains, respectively. Furthermore, GC–MS results represent the average chemical concentration of the entire culture, yet we noticed from the SRS results that the biofuels were highly aggregated as droplets in single cells, which result in a much higher local concentration that facilitates SRS detection.

## Discussion

Raman spectroscopic imaging of living systems has been a grand challenge due to limited speed in spectral acquisition. By the use of picosecond pulses to focus on a single Raman peak, CARS or SRS imaging of living cells has been demonstrated. By chirping two femtosecond pulses and tuning the Raman shift in the time domain, spectral focusing has allowed spectroscopic CARS or SRS imaging. However, the standard implementation of spectral focusing relies on moving a delay stage mechanically. Because of the slow speed in this scheme, hyperspectral CARS or SRS imaging is commonly done in a frame-by-frame manner, and a hyperspectral cube would take a few minutes. Such speed does not allow the study of a dynamic or living system without spectral distortion. For biological cells, fixation is needed, which may cause unwanted biochemical changes inside a cell.

Here, we addressed this difficulty through a novel implementation of SRS spectroscopic imaging that reaches an ultrafast acquisition speed of 20 μs per SRS spectrum. Compared with our previous implementation using a 12-kHz resonant scanner[21], the polygon scanning system not only improves the speed 5-fold but also achieves high-spectral linearity that increases reliability. More importantly, the delay range of our previous configuration was fixed to ~2 ps, which allowed only a moderate degree of chirping and thus reached a spectral resolution of $28 \, cm^{-1}$. In comparison, the tunable delay range of the current scheme enables a much higher degree of chirping, reaching a spectral resolution within $10 \, cm^{-1}$ in the fingerprint region. Currently, the spectral coverage of $200 \, cm^{-1}$ is due to the spectral bandwidth of the laser sources. However, since the delay range is freely tunable, if combined with broadband lasers by fiber amplification[54] or supercontinuum laser sources, the scheme can potentially be used to obtain the entire fingerprint SRS spectrum within 20 μs. In addition, the high-speed delay scanning scheme can be applied to a broad range of modalities requiring a long delay scan, such as transient absorption spectroscopy and impulsive SRS imaging.

In this work, we trained a spatial-spectral residual net as a supervised denoiser that outperformed conventional unsupervised image restoration algorithms. The encoder-decoder structure alleviates the requirement for training data size, which is of great importance for biomedical imaging, given the high cost associated with acquiring training data. Here, fewer than 20 spectroscopic images ($200 \times 200 \times 128$ size for each image) were used as the training set for each application. Compared to the 3D CNN U-net[37], the SS-ResNet has a reduced model complexity. Thus, on the one hand, when the images are very challenging to denoise, it can formulate a deeper network for better performance. On the other hand, if the image can be denoised equally well by the two methods, the reduced size of the model can always make room for more batches in the GPU memory for faster training. Our supervised denoiser can significantly increase

the reliability of the subsequent chemical content analysis. Besides, for the task of denoising spectroscopic image stacks, due to the universal properties of noise under the same imaging conditions, a trained network can be quickly tweaked to denoise other samples by transfer learning. As shown in Supplementary Fig. 13, we applied a network pre-trained on Mia PaCa-2 cells to recover prostate tissue images taken under the same imaging conditions. Direct application achieved high-SNR levels but sacrificed spatial resolution due to the differences between spatial features for the two datasets. By feeding in training data of the new samples, the network required less than half of the training epochs to converge and output high-resolution, high-SNR images, making it convenient to apply to different applications.

It is important to discuss how far the network can push the physical limit of SRS imaging. As discussed in the three demonstrations, the lowest SNR that a network can recover is dependent on the morphological and spectral structures of the samples. In general, the network performs better for images with complex structures, such as cancer cells and tissue. For the ground truth images, the SNR should reach at least ~10 for optimal recovery quality. Since SNR is proportional to the square root of imaging time, we select 100 averages in our case. Further increasing the number of averages for the GT does not improve the denoising quality that much as the bottleneck has become the noise level of the input, whereas decreasing the number of averages will lead to poor quality of recovery since the network cannot learn the actual structures. Thus, the deep-learning network used here can increase the imaging speed by roughly two orders of magnitude.

Like most deep-learning-aided optical imaging applications, the most time-consuming part of the imaging and analysis pipeline is the training of the network. Here, training from the beginning takes ~10 h to finish. However, after training, the network takes only 2 s to process an image. In addition, for similar experimental conditions, a pre-trained network can be quickly adjusted through transfer learning, which can greatly reduce the training time. In comparison, BM4D denoising takes 3 min for a hyperspectral SRS image of the same size. More importantly, SS-ResNet has better denoising performance to allow for a higher image acquisition throughput during the experiment, which is often more critical than the offline processing speed. LASSO-based spectral unmixing takes <1 min to finish for one hyperspectral image. To further improve the offline unmixing throughput for a large dataset, we can simply run multiple instances in parallel or even use several PCs since the spectral unmixing problem for each image is independent.

For all demonstrated applications, the laser powers on the sample were 15 mW for the pump at the wavelength of 891 nm and 75 mW for the Stokes at the wavelength of 1040 nm. An advantage of the SRL modality is that most laser power is on the Stokes beam, which has a longer wavelength and, consequently, a higher damage threshold. Overall, the powers used in our experiments are far below the cell damage threshold of femtosecond pulses at corresponding wavelengths, characterized by our earlier work on SRS microscopy[55,56] and others' work on multiphoton microscopy[57]. Notably, in this work, we applied extensive pulse chirping for both beams, which much reduced the laser peak powers and diminished the nonlinear damage consequently. For the Mia PaCa-2 training and validation samples (Fig. 2a–c), a few very bright spots were observed. These spots were only found occasionally in fixed cells, which are likely the aggregates of cell debris formed during the fixation process. These aggregates were floating in the environment and could easily attach to the cells. In comparison, we did not observe such bright spots in continuous imaging of live Mia PaCa-2 cells (Supplementary Videos 1–5), nor did we found cell membrane blebbing, a signature of cell

membrane damage[55]. Before training, we performed image normalization using 0.3 and 99.7% intensity percentile to avoid including the bright spots. Therefore, these spots did not affect the actual performance of the network.

It is important to discuss whether cell fixation alters the spectral profiles used in our hyperspectral SRS imaging. We used a 4% formalin to fix the cells, which cross-links proteins chemically. It is reported that after cross-linking, the amide I band showed a general peak intensity decrease, but the peak position had no shifting[58]. Comparing SRS images of fixes and live cells, we did not observe significant cellular intensity change to alter the SNR of the raw images. Besides, since we used intensity-normalized spectral profiles as references, we anticipate that cross-linking of proteins does not affect the accuracy of spectral unmixing.

In our setup, we used a low groove density (300 grooves/mm) grating (GR50-0310, Thorlabs) in Littrow configuration as the wedge to introduce optical delay. It is important to know whether the grating can cause angular dispersion to the beam and subsequently induce spatial resolution degradation. As shown in Supplementary Fig. 14, using ×60 1.2 NA water objective, 800 nm pump, and 1040 nm Stokes, we imaged 500-nm PMMA beads to validate the PSF using polygon scanning and standard frame-by-frame SRS. The FWHMs of the beads are 520 and 518 nm, respectively, showing very similar spatial resolution between the two setups. Thus, the spatial resolution of the polygon scanning system is not compromised.

Due to the wavelength difference between pump and Stokes, it is a good practice to chirp them using the rods with different lengths. For the fingerprint region data reported in this work, we used 5 SF57 rods (15-cm each) on the common path and added a 15-cm rod on the Stokes path to compensate for its longer wavelength. For the CH region, the wavelength difference is more significant. Therefore, the spectral resolution using this "5 + 1" configuration leads to $11.7 \, \mathrm{cm}^{-1}$ spectral resolution. We can optimize and push the spectral resolution below $10 \, \mathrm{cm}^{-1}$ by adding more glass rods on the Stokes path to compensate for the increased wavelength difference. Yet, we note that an optimized chirping condition for the CH region will sacrifice the spectral resolution in the fingerprint region due to the over-chirping of Stokes. Therefore, we chose the "5 + 1" chirping condition in this work.

In conclusion, the combination of ultrafast tuning via a polygon scanner and SNR recovery via deep learning has enabled reliable fingerprint SRS imaging at microsecond spectral acquisition speed. The improved speed and spectral resolution by our polygon-based delay tuning of chirped pulses are essential for SRS imaging in the fingerprint region. Meanwhile, the learning network allowed effective SNR enhancement by one order of magnitude. With such advances, we have demonstrated simultaneously imaging of various biomolecules that are difficult to identify in the high-wavenumber C–H window. This technique has broad applications, as demonstrated in this study: from monitoring biofuel production levels in engineered bacteria to the metabolic study of cancer cells, up to large-area whole-brain tissue imaging. Collectively, our approach opens the door to a plethora of biomedical applications from tracking dynamics and interactions of metabolites in a single cell to the high-throughput compositional mapping of an unprocessed human tissue.

## Methods

**Ultrafast tuning spectroscopic SRS microscope**. The ultrafast tuning SRS microscope is illustrated in Supplementary Fig. 1a. A dual-output 80-MHz femtosecond pulsed laser (InSight DeepSee+, Spectra-Physics) outputs synchronized pump and Stokes beams. The 120-fs tunable output (680–1300 nm) was used as the pump beam, while the 200-fs output fixed at 1040 nm served as the Stokes beam.

The Stokes beam was modulated by an acousto-optical modulator (AOM, 1205-C, Isomet) at 2.4 MHz for heterodyne detection. The Stokes beam was then directed to a polygon scanner (Lincoln SA24, Cambridge Technology), which scanned the laser onto a blazed grating (GR50-0310, Thorlabs) positioned at Littrow configuration. The grating acted as a reflective wedge to reflect the Stokes beam along the same optical path. Each scan by the polygon scanner thus introduces a continuous increase of light path for a few millimeters, resulting in a series of continuous temporal delays between the pump and the retroreflected Stokes beam. The maximum delay range of the system is determined by the length of the scan line and the blazed angle of the grating. As shown in Fig. 1b, if we define the length of the laser scan line as $L$ and the grating blazed angle as $\alpha$, then by rotating the grating to change the angle between the scan line and the blazed line (denoted as $\theta$), the effective delay is reduced to $L\sin\theta\tan\alpha$. The beams were collinearly combined by a dichroic mirror (DM, Chroma) and were both broadened to picosecond by high dispersion glass rods (SF57). To compensate for the chirping difference due to the wavelength of the two lasers, we used five 15-cm glass rods on the common path and added one 15-cm rod on the Stokes path. The angle $\theta$ was set to ~20 degrees to match the range of the delay line and the degree of chirping. The chirped beams were sent collinearly to an upright microscope, and a 2D galvo scanner set (GVS102, Thorlabs) was used for scanning images. The pixel size was 200 nm for all applications. A ×60, 1.2 NA water immersion objective (UPLSASP 60XW, Olympus) was used to focus the light onto the sample, followed by forward collection by an oil immersion condenser. For all the experiments, the powers on the samples were 15 mW for the 891 nm pump beam and 75 mW for the 1040 nm Stokes beam. After filtering the Stokes beam following the interaction with the sample, a photodiode (S3994-01, Hamamatsu) with a custom-built resonant circuit was used to collect signals. The SRS signal was extracted by a lock-in amplifier (UHFLI, Zurich Instrument) and was digitized by a high-speed data acquisition card (ATS 460, AlazarTech). A custom-written Matlab (MathWorks) code was used to synchronize the spectral scanning by the polygon scanner and the spatial scanning of the galvo mirrors, generating spectroscopic image stacks in a $\lambda - \mathrm{XY}$ manner.

**Deep-learning network structure, training, and error quantification**. The network for SNR recovery of the spectroscopic image is based on U-net (Supplementary Fig. 3a), which consists of an encoder-decoder architecture. At each layer, a $1 \times 1 \times 1$ convolution layer is first used to increase the feature dimensions, followed by a total of six spatial-spectral convolution (SS-Conv, Supplementary Fig. 3b) layers. A max-pooling layer is applied at the end of each layer to reduce the dimensions. In the decoder phase, each layer first up samples the feature map and then concatenates it with the corresponding feature maps in the encoder phase. The same six SS-Conv layers are used at each layer. At the final stage, a $1 \times 1 \times 1$ convolution layer with linear activation was used to map the feature maps into the prediction of pixel values of the high-SNR image. In addition, the prediction layer was added with the input layer such that the prediction value was the residual[40] with respect to the raw input image, which has been shown to predict higher resolution images. The parameters were learned by minimizing a loss function that averages the mean squared error between the prediction and ground truth. The network was implemented using Keras with Tensorflow as backend and was trained using a graphics processing unit (GPU, RTX 2080 Ti, Nvidia).

To quantify the reconstruction error and compare it with the raw input, we first normalized the ground truth and the predicted image to the same dynamic range by the same method reported in the deep-learning image restoration work[37]. We then calculated the normalized root mean squared error (NRMSE) and structural similarity index (SSIM) using the normalized image pairs. Using the same procedure, we calculated NRMSE and SSIM values between the raw input and ground truth in comparison.

**Linear unmixing of spectroscopic images using pixel-wise LASSO**. Assuming the dimensions of the spectroscopic image in $x, y, \lambda$ as $N_x, N_y, N_\lambda$, we first rearrange the 3D spectroscopic image stack as a 2D data matrix ($D \in \mathbb{R}^{N_x N_y \times N_\lambda}$) by arranging the pixels in the raster order. Given the number of pure components as $K$, a bilinear model is used to decompose the data matrix into the multiplication of concentration maps $C \in \mathbb{R}^{N_x N_y \times K}$ and spectral profiles of pure chemicals $S \in \mathbb{R}^{K \times N_\lambda}$:

$$D = CS + E \qquad (1)$$

where $E$ is the residual term. To simplify the problem, we obtained $S$ by measuring the spectral profiles from pure chemicals. The concentrations can be obtained by minimizing the error term $E$ through the least-squares fitting. However, in practice, least-squares fitting alone generates chemical maps with severe crosstalks in complex biological samples where many biochemicals have overlapping spectral profiles. To improve the performance, we observe that for each spatial pixel, only a few chemical components contribute significantly, which is equivalent to the sparsity of concentrations at each pixel. Thus, we introduced an $L_1$-norm regularization to the original least-squares fitting problem. We denote $i = 1, \ldots, N_x N_y$ as a spatial pixel location and formulate the following optimization

problem:

$$\hat{C}_i = \arg\min_{C_i}\{\tfrac{1}{2}\|D(i,:) - C_i S\|^2 + \beta\|C_i\|_1\} \qquad (2)$$

where $\beta$ is a hyper-parameter controlling the level of the sparsity of the concentration, $D(i,:) \in \mathbb{R}^{N_\lambda}$ stands for the spectrum at pixel $i$ and $C_i \in \mathbb{R}^K$ is a vector that contains all the concentration values at the pixel. For a set of data recorded in the same imaging and digitizing conditions, the value of $\beta$ needs tuning only once. The method, known as the least absolute shrinkage and selection operator (LASSO), has been widely used to solve problems in which the variable is sparse, e.g., compressed sensing[59]. With the use of pixel-wise LASSO unmixing, it is possible to resolve more chemicals in the same window since LASSO effectively stabilizes the solution and suppresses the crosstalks between different channels, especially for a complex living system when the spectral profiles of independent chemical components are similar. The method does not have a strict constraint on the level of sparsity or the maximum number of independent chemical components to tolerate, it is a soft regularization method, and the levels of sparsity can tone down in the case of multiple mixtures.

**Mia PaCa-2 cancer cell preparation**. Mia PaCa-2 cancer cells were grown in a monolayer at 37 °C in 5% $CO_2$ in RPMI-1640 medium supplemented with 10% fetal bovine serum. To prepare fixed cell samples for training, we cultured Mia PaCa-2 cells on a glass-bottom dish for 1–2 days at the humidified chamber and fixed them with 10% neutral buffered formalin for 15 min at room temperature. The cells were then washed with and imaged in PBS buffer. For cholesterol depletion in Mia PaCa-2 cells, 500 μM HPβCD was added to the medium and cultured for 24 h.

**Brain tissue preparation**. The mouse brain slice was prepared from a mouse (Jackson Lab) at age 21 days. The housing conditions were dark/light 12/12 light cycle, 21 ± 3 °C temperature, and between 30 and 70% humidity. PBS was used for perfusion, after which formalin was perfused to fix the brain tissue. Then the brain tissue was frozen-sectioned at 150-μm thickness.

**E. coli biofuel strains**. The E. coli strains used in this study are derived from strain JW0451-2 (K-12 BW25113 ΔacrB) from the Keio collection[60]. The kanamycin resistance marker gene was removed from the Keio collection strain. This "wild type" strain was then transformed with plasmids expressing the heterologous pathways for either pinene or limonene production. For pinene production, the chassis strain was transformed with two plasmids, pJBEI-3933 & pJBEI-3085[51], which were gifts from Jay D. Keasling. For limonene production, the chassis strain was transformed with plasmid pJBEI-6409[50], provided by Taek Soon Lee via Addgene (#47048).

Prior to SRS imaging, overnight cultures were inoculated in Luria Bertani (LB) medium with appropriate antibiotics for plasmid maintenance and refreshed the following day in 5 mL of M9 minimal media supplemented with 20 g/L glucose and appropriate antibiotics. When the cultures reached an $OD_{600}$ (optical density at 600 nm) of 0.6, pinene or limonene production was induced by adding IPTG to the culture (500 and 25 μM, respectively). The cultures were grown at 37 °C for another 18–24 h. Five to ten minutes before imaging, 5 μL of culture was placed on a 3% agarose pad and pressed between microscope coverslips to immobilize the cells, and then the sample was imaged.

**Statistics and reproducibility**. For each demonstration, the SS-ResNet was independently trained three times with similar denoising and spectral unmixing results. For Fig. 2 and Supplementary Fig. 5, 7 technical replicates of the Mia PaCa-2 cells with both control and drug treatment were obtained. For Fig. 3, Supplementary Figs. 6 and 8, SRS imaging of represented brain tissue regions was repeated three times. For Fig. 4 and Supplementary Fig. 11, 4 technical replicates of all the demonstrated E. coli strains were measured. For Supplementary Fig. 13, the transfer learning results were repeated using prostate cancer tissue images on 2 different days. For Supplementary Fig. 14, the spatial resolution calculation was independently repeated in three different field-of-views.

**Reporting summary**. Further information on research design is available in the Nature Research Reporting Summary linked to this article.

## Data availability

All the data related to the work is available upon reasonable request to the corresponding author. Example datasets for neural network training and spectral unmixing are available on the following website: https://github.com/buchenglab.

## Code availability

Code for SS-ResNet and LASSO spectral unmixing is available on the following website: https://github.com/buchenglab.

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

## Acknowledgements

This work is supported by a DOE grant BER DE-SC0019387 to M.J.D., W.W., and J.X.C. and NIH grants R01GM118471, R33CA223581, and R01AI141439 to J.X.C.

## Author contributions

H.L., C.Z. and F.D. designed and implemented the ultrafast tuning system. H.L. wrote the code for SS-ResNet SNR recovery and LASSO spectral unmixing. H.L. and H.J.L. performed experiments on Mia PaCa-2 cells and mouse brain. H.L., N.T., and J.B.L. performed experiments on *E. coli* biofuel production. J.S. provided GC–MS results for biofuel production. H.L., H.J.L. and J.X.C. wrote the manuscript with input from M.J.D., N.T., L.T. and J.B.L.

## Competing interests

The authors declare no competing interests.
