## [Peer Review File · Nature Communications]

Reviewers' Comments:

Reviewer #1:

Remarks to the Author:

There are no significant advances in Machine Learning presented here. This paper therefore should be judged in terms of its contributions to nonlinear microscopy.

I am not convinced that 'spatial-spectral learning' needs to be the title. It seems like just jumping on the highly fashionable Machine Learning bandwagon.

Can the authors explain what is their major advance in Machine Learning? If there isn't one (I don't see it) then it should not be emphasized in the title.

The authors have done a lot of work here and presented some very nice data and impressive images.

To be critical, I could say that to a large degree, these authors have copied the polygon scanner idea used for FT-CARS (their reference 30) and applied ML to improve the S/N ratio.

Before a publication decision can be made, the authors should answer the questions/concerns below.

[1] The spectral resolution claim is based on a measurement at 720 cm^{-1} which is taken with the pump and Stokes having relatively close center wavelengths. However, for CH imaging, the center wavelengths are significantly different and higher order dispersion may play a role. For example, this paper: [Mojtaba Mohseni, Christoph Polzer, and Thomas Hellerer, "Resolution of spectral focusing in coherent Raman imaging," *Opt. Express* 26, 10230-10241 (2018)] has found that higher order chirp limits resolution in the CH region. What spectral resolution do the authors achieve in the CH region?

[2] Related to the above question, in the supplementary data, it would be helpful if Fig. 2(d) and (f) had the same axes so they can be more easily compared. It would also be good to have all of 2(c) replotted with a wavenumber axis after calibration. Based on the measurements in the supplementary figure, what is the spectral resolution if you use the main peak of DMSO?

[3] I would expect the beam from the grating to have angular dispersion. This should change the pulse length as a function of propagation distance and possibly change the spatial mode at the focus, leading to spatial spectral coupling. How does the point spread function of this setup compare to a more typical system? In the FT-CARS paper cited in the manuscript (ref 30), 15 ps of group delay was achieved using a 4f setup which does not introduce spatial chirp and angular dispersion, so it is somewhat unclear why this new geometry was chosen.

[4] For biofuel production, the authors state they performed quantitative imaging of limonene and pinene; however, there does not seem to be independent verification of quantities measured.

[5] I would also point out that two of their demo systems don't really need the experimental set up they built. For the brain imaging, they are looking at fixed slices. Yes they can go faster, but it doesn't seem necessary. It wasn't clear to me what the advantage is over current techniques that just look at a few CH bands. For the biofuels, the main trick seems to be doing linear unmixing. Maybe it needs to be fast so you can image live e-coli I guess, but the unmixing seems to be the key. Can the authors comment?

This paper should be reviewed again after the revisions.

Reviewer #2:

Remarks to the Author:

In this manuscript, Lin et.al developed a method of ultrafast spectroscopic SRS imaging with combined fast delay-line scanning and deep-learning network. The method achieved both denoising with improved spatial quality and high spectral fidelity with improved chemical resolution. Although the polygon based delay scanner is no longer new, the major merit of the work exist in that the integrated imaging speed and spectral information has been pushed to a new forefront. The work is carefully done with high quality data, I would suggest its publication with the following minor revisions:

1. When discussing the limitation of previous delay scanning techniques, the authors seemed to have missed the design used in time-domain OCT, which was also demonstrated for spectral focusing SRS [Optics Letters 2017, 42 (4), 659-662] with large delay tuning range and high linearity.
2. The size of the training dataset used in SS-ResNet is very small (less than 20). While this might be sufficient for cell imaging, I would expect much larger datasize for tissue imaging, given the more complex spatial patterns in tissues. What are the sizes for different applications?
3. The SRS spectra produced by SS-ResNet seem rather smooth, a comparison from the ground truth spectra would be helpful.
4. How well does the trained SS-ResNet work for untrained chemical species in the same spectral range? The authors may want to experiment on this and show the results.
5. Citation of ref. 41 needs to be corrected.

Reviewer #3:

Remarks to the Author:

Lin et al presents a novel spectral sampling methodology combined with a deep learning data post-treatment to enable chemically specific imaging with higher selectivity than previous work. For a long time, spectral imaging of the so-called molecular fingerprint was known to be the most powerful approach in terms of chemical selectivity, but extremely difficult to achieve at high speed because the signal levels are too weak. Accessing such spectral region would allow for imaging at a much higher chemical selectivity than conventional SRS microscopy does (typically focused on the boring, and limited to 2-3 species, carbon-hydrogen stretch congested region). The main point of the paper is to show that a synergetic combination of novel ultrafast sampling scheme with deep learning enables an effective methodology for high-speed vibrational fingerprint imaging. In effect, they show that such high speed imaging is imperative, otherwise motion artefacts would complicate analysis. Various standard specimens in SRS microscopy have been exemplified using their methods.

On the spectral sampling method, the authors have developed a brand new approach (polygon sampling with the grating) that allowed for highly linear sampling, therefore avoiding spectral distortions. On the deep learning side, they smartly adapted a well-known network (U-Net type), used for the 3-D imaging, into the spectral imaging problem. What is the most exciting in their approach is the rather simple training of the network with only a handful of examples in the training step: this is by no means trivial in spectral imaging as standard deep learning methodologies requires prohibitive training set sizes ($10^3 - 10^4$) that are hard to obtain in SRS spectral imaging. All these features together certainly merit the work publication in Nature Communications. However, I believe they still need to address a few, yet important, missing information to put their methodology in perspective for high throughput SRS spectral imaging. A list of points to address in a new version is given below.

Major points:

- 1) What's the physical insight into what is the network learning? I missed this in the discussion. Is it some sort of spatial-spectral correlation (the noise is isotropic)? Related to this point, is there any effect of spatial oversampling on the SSIM levels? (speaking of which, the authors don't

mention the level of oversampling (psf size/ pixel size)).

2) Although there is much originality of the polygon sampling with the new network design, I don't see much novelty on the use of the LASSO scheme (it has been used extensively in the past), so I don't understand why the authors emphasise it so much. Related to that, why the authors used supervised methods? The power of fingerprint imaging is that the spectra are very different, so I would expect NMF methods to work very well (specially under such high SNR settings) after denoising by the network. Did the authors compared the NMF with the supervised method? Another issue with L1 optimisation problems is reconstruction time (see #3 below).

3) There is a big claim (in various parts of the manuscript) that their methodology would increase image throughput, whereas I could not find any information on absolute timings taken for the calculations (or at least discussions). It is clear that the methodology allows for faster acquisition, but it is not proven that the whole pipeline is actually improving the analysis throughput. The authors should state the time taken for training and reconstruction (SS-ResNet and LASSO), and how does that compare with the standard methods they compare with in the paper.

4) I appreciated the fact that the authors don't oversimplify the Deep Learning issues (the problem of training set size), but they should be more explicit on the problems that still exists, for instance, on the universality of the network for imaging other specimens. Transfer learning is an interesting solution, but the results are only modest as the images clearly show artefacts (blurry motions, low resolution) comparable to the motion artefacts mentioned by the authors in other parts of the text. It would be good to see the SSIM analysis of such approach.

Minor points:

1) It would be good to show in the supplemental, a comparison between the LASSO and a Tikhonov regulariser, or at least the reasoning why it is not ideal.

2) In the introduction (51-53), the authors (too) briefly introduced the different ways of performing spectroscopic imaging. It looks like they focused mostly on their own achievements leaving aside a number of original work that is still discussible if they wouldn't provide faster imaging speeds, e.g. like 10.1364/OE.21.015113, 10.1364/OL.42.000294, 10.1364/OE.23.025235, 10.1364/OL.41.003021, 10.1364/OL.42.001696

This point is seriously overlooked throughout the manuscript and is further repeated in lines 389-390 with outdated information (it is well known that delay stage is incompatible with high speed imaging, that's why acousto-optical methods have been developed).

3) Why the SSIM index does not reach 1 in the various examples?

4) line 537: is E error or noise?

Point-by-point response to reviewers' comments:

REVIEWER COMMENTS

Reviewer #1 (Remarks to the Author):

There are no significant advances in Machine Learning presented here. This paper therefore should be judged in terms of its contributions to nonlinear microscopy. I am not convinced that 'spatial-spectral learning' needs to be the title. It seems like just jumping on the highly fashionable Machine Learning bandwagon.

Can the authors explain what is their major advance in Machine Learning? If there isn't one (I don't see it) then it should not be emphasized in the title.

Response: We appreciate the comments by the reviewer and would like to explain the novelty of our deep learning approach. Here, we handle hyperspectral SRS images that are 3D data cubes (2D spatial + 1D spectral) yet differ from volumetric image data. Therefore, the noise properties are inhomogeneous, and the physical correlations are dissimilar in the spatial and spectral dimensions. Inspired by the approaches in the video-processing community (ref. 38), we modified the convolution kernel from a 3D CNN to a 2+1 spatial-spectral convolution kernel and adapted it into a U-net residual learning network. The spatial-spectral kernel has fewer trainable parameters and thus reduces the model complexity. On the one hand, when the images are very challenging to denoise, it can formulate a deeper network for better performance. On the other hand, if the image can be denoised equally well by the two methods, the reduced size of the model can always make room for more batches in the GPU memory for faster training.

The main goal of this work is to integrate advanced instrumentation with data science approach to push the physical limits of SRS microscopy to a new forefront. Here, we achieved unprecedented microsecond-level fingerprint spectroscopic SRS imaging with high data fidelity, which is possible only by the joint efforts in advanced instrumentation and deep learning. Therefore, we would like to emphasize both points in the title.

The authors have done a lot of work here and presented some very nice data and impressive images.

Response: We thank the reviewer for the positive opinion on our efforts towards pushing the boundaries of SRS imaging and the production of high-quality data.

To be critical, I could say that to a large degree, these authors have copied the polygon scanner idea used for FT-CARS (their reference 30) and applied ML to improve the S/N ratio.

Response: We appreciate this critical comment and would like to point out the difference between our polygon scanner and the one used for FT-CARS. As discussed in the manuscript's introduction, our setup has two key innovations that significantly improve the versatility and reliability. Firstly, our scheme maintains very high linearity between the data sampling from the trigger and corresponding Raman shifts, making the recorded spectrum free of spectral stretching distortion. Secondly, by rotating the blazed grating to change the angle between the laser-scanned line and the blazed lines of the grating (angle θ in

Fig. 1b), the maximum delay is tunable, which allows a perfect match between the delay range and an arbitrary degree of pulse chirping.

Before a publication decision can be made, the authors should answer the questions/concerns below.

[1] The spectral resolution claim is based on a measurement at 720 cm⁻¹ which is taken with the pump and Stokes having relatively close center wavelengths. However, for CH imaging, the center wavelengths are significantly different and higher order dispersion may play a role. For example, this paper: [Mojtaba Mohseni, Christoph Polzer, and Thomas Hellerer, "Resolution of spectral focusing in coherent Raman imaging," Opt. Express 26, 10230-10241 (2018)] has found that higher order chirp limits resolution in the CH region. What spectral resolution do the authors achieve in the CH region? What spectral resolution do the authors achieve in the CH region?

Response: We appreciate the reviewer for bringing up this very important point. We have detailed the description of glass rods arrangements in the Methods section of the manuscript. Indeed, due to the wavelength difference between pump and Stokes, it is a good practice to chirp them using the rods with different lengths. For the fingerprint region data reported in this work, we used 5 SF-57 rods (15-cm each) on the common path and added a 15-cm rod on the Stokes path to compensate for its longer wavelength.

For the CH region, as pointed out by the reviewer, the wavelength difference is more significant. Therefore, the spectral resolution using this "5+1" configuration leads to 11.7 cm⁻¹ spectral resolution. We can optimize and push the spectral resolution below 10 cm⁻¹ by adding more glass rods on the Stokes path to compensate for the increased wavelength difference. Yet, we note that an optimized chirping condition for the CH region will sacrifice the spectral resolution in the fingerprint region due to the over-chirping of Stokes. Therefore, we chose the "5+1" chirping condition in this work.

We have attached a comparison of the spectral resolution using two different chirping configurations in the response letter below. The top row demonstrates the spectral resolution in the CH region by 5+2 and 5+1 chirping, while the bottom row shows the results in the fingerprint region, respectively.

[2] Related to the above question, in the supplementary data, it would be helpful if Fig. 2(d) and (f) had the same axes so they can be more easily compared. It would also be good to have all of 2(c) replotted with a wavenumber axis after calibration. Based on the measurements in the supplementary figure, what is the spectral resolution if you use the main peak of DMSO?

Response: We thank the reviewer for the suggestion and modified the supplementary figure accordingly. The FWHM of the DMSO main peak is 16.4 cm^{-1} with the 5+1 rods setup. After deconvolution with the peak width in spontaneous Raman (11.5 cm^{-1}), we can reach a spectral resolution of 11.7 cm^{-1} .

[3] I would expect the beam from the grating to have angular dispersion. This should change the pulse length as a function of propagation distance and possibly change the spatial mode at the focus, leading to spatial spectral coupling. How does the point spread function of this setup compare to a more typical system? In the FT-CARS paper cited in the manuscript (ref 30), 15 ps of group delay was achieved using a 4f setup which does not introduce spatial chirp and angular dispersion, so it is somewhat unclear why this new geometry was chosen.

Response: We thank the reviewer for this very good insight. In our setup, we used a low groove density (300 grooves/mm) grating (GR50-0310, Thorlabs) in Littrow-configuration to reflect a focused Stokes beam. From the experimental tests, the angular dispersion is not significant at the focus to induce significant distortion of spatial resolution. As shown below, using 60X 1.2 NA water objective, 800nm pump, and 1040 nm Stokes, we imaged 500-nm PMMA beads to validate the PSF using polygon scanning and standard frame-by-frame. The FWHMs of the beads are 520 nm and 518 nm, respectively, showing very similar spatial resolution between the two setups. We have added the results as Supplementary Fig. 13.

As we described in the introduction, compared to the one in the FT-CARS paper, our first advantage is the spectral linearity of the data, making data acquisition, processing, and analysis much more robust. The second major advantage is the tunable delay range of the setup, which can guarantee a perfect match between any degree of chirping and scanned delay range to maximize data fidelity. The tunable delay

range also makes it useful for other pump-probe modalities requiring various delay tuning range, such as transient absorption spectroscopic imaging, impulsive SRS, etc.

[4] For biofuel production, the authors state they performed quantitative imaging of limonene and pinene; however, there does not seem to be independent verification of quantities measured.

Response: We appreciate the reviewer's comment on this very important point. The demonstrated limonene and pinene producing strains were previously reported in the literature (refs 50 & 51). As an independent validation, we have performed GCMS of the wild type, limonene, and pinene producing strains shown in the main figure. From the GCMS results, limonene and pinene peaks clearly present in the limonene and pinene production strains. We note that GCMS results represent the average chemical concentration of the entire culture. Yet, from the SRS results, the biofuels were highly aggregated as droplets in single cells, which result in a much higher local concentration that facilitates SRS detection. The GCMS results have been added to the manuscript as supplementary information.

Indeed, obtaining absolute chemical concentrations based on SRS measurements is difficult. However, as SRS intensity is linear with the molecular concentration, a quantitative comparison of the biofuel concentrations between cells and among different strains are feasible. Since our goals are to detect single-cell production level heterogeneity and screen the production levels between different strains, such semi-quantitative measurements that represent relative concentration differences are sufficient for selection and screening.

[5] I would also point out that two of their demo systems don't really need the experimental set up they built. For the brain imaging, they are looking at fixed slices. Yes they can go faster, but it doesn't seem necessary. It wasn't clear to me what the advantage is over current techniques that just look at a few CH bands. For the biofuels, the main trick seems to be doing linear unmixing. Maybe it needs to be fast so you can image live e-coli I guess, but the unmixing seems to be the key. Can the authors comment?

Response: We appreciate the comment and would justify the necessity of using our approach. Here, the fingerprint vibration bands are essential to map lipid unsaturation and cholesterol and to map the biofuels in single bacteria.

For brain imaging, although the brain slice was fixed, the size was very large (0.5 cm*1 cm). For such a large-scale sample, it is very difficult to obtain hyperspectral SRS in the fingerprint region within a manageable time. In this work, the raw brain data (high-speed, low-SNR) already took 3.5 hours to finish.

Without the high-throughput ability of our approach, acquisition of high-quality data would need 100 times average, which would take $100 \times 3.5 \text{ h} = 14$ days to finish.

For the *E. coli* biofuel production project, high speed is necessary for the following two reasons. First, the final goal is to screen efficient biofuel producers over a complicated library with hundreds of strains, and a high-throughput SRS imaging method is crucial for efficient screening. Second, as commented by the reviewer, our platform enables live *E. coli* imaging, making it possible to collect the high producers within the same strain and regrow them to reach a higher production yield.

This paper should be reviewed again after the revisions.

Reviewer #2 (Remarks to the Author):

In this manuscript, Lin et al. developed a method of ultrafast spectroscopic SRS imaging with combined fast delay-line scanning and deep-learning network. The method achieved both denoising with improved spatial quality and high spectral fidelity with improved chemical resolution. Although the polygon based delay scanner is no longer new, the major merit of the work exist in that the integrated imaging speed and spectral information has been pushed to a new forefront. The work is carefully done with high quality data, I would suggest its publication with the following minor revisions:

Response: We appreciate the positive comments from the reviewer.

1. When discussing the limitation of previous delay scanning techniques, the authors seemed to have missed the design used in time-domain OCT, which was also demonstrated for spectral focusing SRS [Optics Letters 2017, 42 (4), 659-662] with large delay tuning range and high linearity.

Response: We appreciate the reviewer for pointing out the paper. We have added it to the introduction section.

2. The size of the training dataset used in SS-ResNet is very small (less than 20). While this might be sufficient for cell imaging, I would expect much larger datasize for tissue imaging, given the more complex spatial patterns in tissues. What are the sizes for different applications?

Response: We apologize for the confusion. For MiaPaCa-2 and *E. coli* imaging, the training dataset was less than 20 images, each with $(200 \times 200 \times 128)$ in size. For the brain tissue imaging, the training set consists of images taken at eight different brain regions to make sure different spatial features can be learned by the network. The training images at various regions were stitched by several smaller ROIs, the total number of training images is ~ 50 in terms of the small images. We have added the description of the training data size to the main text.

3. The SRS spectra produced by SS-ResNet seem rather smooth, a comparison from the ground truth spectra would be helpful.

Response: We appreciate the suggestion from the reviewer. Single-pixel spectrum from raw, ground truth, and the network output for all the three applications are included in the Supplementary Figs. 5, 8, 11.

4. How well does the trained SS-ResNet work for untrained chemical species in the same spectral range? The authors may want to experiment on this and show the results.

Response: During the training process, a deep learning denoiser mainly takes advantage of the spatial and spectral correlations of cells. For the same spectral window and the same group of cells with similar SNR, denoising a dataset with an untrained chemical species should work since different chemical components do not alter the overall image SNR and cell morphologies.

To validate this, we used a trained network using the wild type, limonene-producing and pinene-producing strains and applied it to a fatty-acid producing cell strain acquired on the same day. As demonstrated in the first two applications, fatty acid has a strong C=C peak at 1650 cm^{-1} . As shown in the figures below, the quality of the denoised image is comparable to the other three strains demonstrated in the main text. Subsequent spectral unmixing can also produce the untrained fatty acid chemical map.

5. Citation of ref. 41 needs to be corrected.

Response: We apologize for the mistake, the reference has been corrected, thank you!

Reviewer #3 (Remarks to the Author):

Lin et al presents a novel spectral sampling methodology combined with a deep learning data post-treatment to enable chemically specific imaging with higher selectivity than previous work. For a long time, spectral imaging of the so-called molecular fingerprint was known to be the most powerful approach in terms of chemical selectivity, but extremely difficult to achieve at high speed because the signal levels are too weak. Accessing such spectral region would allow for imaging at a much higher chemical selectivity than conventional SRS microscopy does (typically focused on the boring, and limited to 2-3 species, carbon-hydrogen stretch congested region). The main point of the paper is to show that a synergetic combination of novel ultrafast sampling scheme with deep learning enables an effective methodology for high-speed vibrational fingerprint imaging. In effect, they show that such high speed

imaging is imperative, otherwise motion artefacts would complicate analysis. Various standard specimens in SRS microscopy have been exemplified using their methods.

On the spectral sampling method, the authors have developed a brand new approach (polygon sampling with the grating) that allowed for highly linear sampling, therefore avoiding spectral distortions. On the deep learning side, they smartly adapted a well-known network (U-Net type), used for the 3-D imaging, into the spectral imaging problem. What is the most exciting in their approach is the rather simple training of the network with only a handful of examples in the training step: this is by no means trivial in spectral imaging as standard deep learning methodologies requires prohibitive training set sizes ($10^3 - 10^4$) that are hard to obtain in SRS spectral imaging. All these features together certainly merit the work publication in Nature Communications. However, I believe they still need to address a few, yet important, missing information to put their methodology in perspective for high throughput SRS spectral imaging. A list of points to address in a new version is given below.

Response: We deeply appreciate the positive feedback from the reviewer.

Major points:

1) What's the physical insight into what is the network learning? I missed this in the discussion. Is it some sort of spatial-spectral correlation (the noise is isotropic)? Related to this point, is there any effect of spatial oversampling on the SSIM levels? (speaking of which, the authors don't mention the level of oversampling (psf size/ pixel size)).

Response: Given the nature of CNN, the network is mainly learning spatial and spectral correlation features and noise levels. Since the images were obtained in $\lambda - XY$ manner, the noise is indeed isotropic, and the spectral noise is lower than the frame-by-frame approach due to much-reduced low-frequency noise.

The pixel size in our applications is 200 nm. The FWHM of the SRS PSF is ~ 290 nm based on the equation $\frac{0.61\lambda}{\sqrt{2} NA}$. We have added the description of pixel size to the Methods section.

2) Although there is much originality of the polygon sampling with the new network design, I don't see much novelty on the use of the LASSO scheme (it has been used extensively in the past), so I don't understand why the authors emphasise it so much. Related to that, why the authors used supervised methods? The power of fingerprint imaging is that the spectra are very different, so I would expect NMF methods to work very well (specially under such high SNR settings) after denoising by the network. Did the authors compared the NMF with the supervised method? Another issue with L1 optimisation problems is reconstruction time (see #3 below).

Response: We appreciate the review for bringing up this excellent point. LASSO has been widely adopted in many fields. However, it has not been used for solving the problem of spectral unmixing of hyperspectral SRS. When processing biological samples, we found that the non-negative matrix factorization (NMF) method (with alternating least square update) often converges to an unstable output as the update for spectral profiles leads to pure components that are not chemically accurate.

To improve the stability of spectral unmixing, we decided to fix the spectral profiles using pure chemicals. We also observed that at each pixel, a few components have dominant contributions. Such physical

conditions can be adapted as a pixel-wise chemical sparsity constraint (L1 norm) in spectral unmixing. For the datasets in this manuscript, protein's broad 1650 cm^{-1} Amide-I peak has spectral overlaps with many other chemicals. Therefore, adding local sparsity constraint is very effective.

We used the standard LASSO core function (ADMM-based) provided in Matlab to solve the unmixing problem. Processing a $200 \times 200 \times 128$ image with 3 to 4 spectral components takes less than a minute to finish on a personal laptop. Additionally, since we are fixing the spectral profiles, LASSO only needs to run once, which is much faster than the alternating least square methods (MCR-ALS), which typically requires many iterations.

3) There is a big claim (in various parts of the manuscript) that their methodology would increase image throughput, whereas I could not find any information on absolute timings taken for the calculations (or at least discussions). It is clear that the methodology allows for faster acquisition, but it is not proven that the whole pipeline is actually improving the analysis throughput. The authors should state the time taken for training and reconstruction (SS-ResNet and LASSO), and how does that compare with the standard methods they compare with in the paper.

Response: We have added the analysis of the throughput for the whole pipeline in the discussion. Like most DL-aided optical imaging applications, the most time-consuming part of the pipeline is the training of the network. Here, training from the beginning takes ~ 10 hours to finish. However, after training, the network takes only 2 seconds to process an image. Additionally, for similar experimental conditions, a pre-trained network can be quickly adjusted by feeding new data (i.e., transfer learning), which can greatly reduce the training time. In comparison, BM4D denoising (Matlab version) takes 3 min for the same hyperspectral SRS image. More importantly, SS-ResNet has better denoising performance to allow for a higher image acquisition throughput during the experiment, which is often more critical than the offline processing speed.

LASSO-based spectral unmixing, as mentioned previously, takes less than 1 min to finish for one hyperspectral image ($200 \times 200 \times 128$) with Matlab's built-in LASSO core function. To further improve the offline unmixing throughput for a large dataset, we can simply run multiple instances in parallel or even use several PCs since the spectral unmixing problem for each image is independent.

4) I appreciated the fact that the authors don't oversimplify the Deep Learning issues (the problem of training set size), but they should be more explicit on the problems that still exists, for instance, on the universality of the network for imaging other specimens. Transfer learning is an interesting solution, but the results are only modest as the images clearly show artefacts (blurry motions, low resolution) comparable to the motion artefacts mentioned by the authors in other parts of the text. It would be good to see the SSIM analysis of such approach.

Response: We appreciate the suggestions from the reviewer. For the three demonstrated applications, the samples are significantly different in terms of the sample dimensions and morphology. Since sample dimensions affect the SRS signal levels and the morphologies are important to the network for denoising, one network cannot work well across all the applications in the manuscript. Thus, we trained three individual networks in the manuscript. As we have mentioned in the discussion, one way to alleviate the training time is through transfer learning, which can shorten the training time, but it is still necessary to obtain a set of new training data each time. We have added SSIM analysis for the transfer learning results in the supplementary information.

Minor points:

1) It would be good to show in the supplemental, a comparison between the LASSO and a Tikhonov regulariser, or at least the reasoning why it is not ideal.

Response: Both LASSO and Tikhonov regularization can facilitate the robustness of least square fitting. However, LASSO can promote a sparser solution, which is closer to the physical prior knowledge that at each pixel, only a few components dominant. In addition, LASSO is more resistant to outliers in the data, which is very useful for suppressing noise or artifacts in the data. These two factors together lead to cleaner and better-separated chemical maps. Further study of the optimal spectral unmixing methods and regularizations under sophisticated conditions, including the noise levels, the number of chemical species, and the number of sampled spectral frames is an ongoing project in our lab and is beyond the scope of this manuscript.

2) In the introduction (51-53), the authors (too) briefly introduced the different ways of performing spectroscopic imaging. It looks like they focused mostly on their own achievements leaving aside a number of original work that is still discussible if they wouldn't provide faster imaging speeds, e.g. like 10.1364/OE.21.015113, 10.1364/OL.42.000294, 10.1364/OE.23.025235, 10.1364/OL.41.003021, 10.1364/OL.42.001696. This point is seriously overlooked throughout the manuscript and is further repeated in lines 389-390 with outdated information (it is well known that delay stage is incompatible with high speed imaging, that's why acousto-optical methods have been developed).

Response: We appreciate the reviewer for suggesting these works. We have added them and modified the introduction. Thank you!

3) Why the SSIM index does not reach 1 in the various examples?

Response: In the work, we use SSIM as a measure to quantify the quality of reconstruction and unmixing quality. An SSIM value of 1 refers to identical images, which is impossible for any denoiser to achieve in practice. The absolute values of SSIM can vary significantly based on the cell types in the image. Thus, for the various examples, measure the denoising and unmixing quality based on both quantitative measures (SSIM and NRMSE) and inspections of the outputs to check whether obvious artifacts exist.

4) line 537: is E error or noise?

Response: E is the noise term introduced during the measurement.

Reviewers' Comments:

Reviewer #1:

Remarks to the Author:

The authors have done a thorough and convincing job in replying to the various referees' comments. I commend them for this and can recommend this paper for acceptance.

Reviewer #2:

Remarks to the Author:

The authors have carefully addressed all my concerns, and I would like to suggest its publication in Nature Communications.

Reviewer #3:

Remarks to the Author:

The authors present a new revised version of the manuscript. I believe this new version has considerably improved, and exposed several points that were not stated in the previous version. The authors have addressed all major criticism I raised by explaining better the details in the main manuscript. To conclude, this paper is a synergistic combination of optical engineering (new spectral sampling scheme) and computational methods (deep neural net) that allowed to expand the applications of SRS fingerprint spectral bio-imaging at high-speed acquisition. Doing each of these approaches separately, optical engineering or computational methods, would not have significantly surpassed the current limits of chemical imaging and, as a matter of fact, would not have significant novelty. Therefore, I strongly recommend publication in Nature Communications.

Point-by-point response to reviewers' comments:

REVIEWER COMMENTS

Reviewer #1 (Remarks to the Author):

The authors have done a thorough and convincing job in replying to the various referees' comments. I commend them for this and can recommend this paper for acceptance.

Response: We deeply appreciate the positive feedback from the reviewer.

Reviewer #2 (Remarks to the Author):

The authors have carefully addressed all my concerns, and I would like to suggest its publication in Nature Communications.

Response: We deeply appreciate the positive feedback from the reviewer.

Reviewer #3 (Remarks to the Author):

The authors present a new revised version of the manuscript. I believe this new version has considerably improved, and exposed several points that were not stated in the previous version. The authors have addressed all major criticism I raised by explaining better the details in the main manuscript. To conclude, this paper is a synergistic combination of optical engineering (new spectral sampling scheme) and computational methods (deep neural net) that allowed to expand the applications of SRS fingerprint spectral bio-imaging at high-speed acquisition. Doing each of these approaches separately, optical engineering or computational methods, would not have significantly surpassed the current limits of chemical imaging and, as a matter of fact, would not have significant novelty. Therefore, I strongly recommend publication in Nature Communications.

Response: We deeply appreciate the positive feedback from the reviewer.